# Tuning the effective spin-orbit coupling in molecular semiconductors

Sam Schott[1], Erik R. McNellis[2], Christian B. Nielsen[3,4], Hung-Yang Chen[3], Shun Watanabe[5,6], Hisaaki Tanaka[7], Iain McCulloch[3,8], Kazuo Takimiya[9], Jairo Sinova[2] & Henning Sirringhaus[1]

The control of spins and spin to charge conversion in organics requires understanding the molecular spin-orbit coupling (SOC), and a means to tune its strength. However, quantifying SOC strengths indirectly through spin relaxation effects has proven difficult due to competing relaxation mechanisms. Here we present a systematic study of the $g$-tensor shift in molecular semiconductors and link it directly to the SOC strength in a series of high-mobility molecular semiconductors with strong potential for future devices. The results demonstrate a rich variability of the molecular $g$-shifts with the effective SOC, depending on subtle aspects of molecular composition and structure. We correlate the above $g$-shifts to spin-lattice relaxation times over four orders of magnitude, from 200 to 0.15 µs, for isolated molecules in solution and relate our findings for isolated molecules in solution to the spin relaxation mechanisms that are likely to be relevant in solid state systems.

[1] Cavendish Laboratory, University of Cambridge, Cambridge CB3 0HE, UK. [2] Institute of Physics, Johannes Gutenberg-Universität, 55128 Mainz, Germany. [3] Department of Chemistry and Centre for Plastic Electronics, Imperial College London, London SW7 2AZ, UK. [4] Materials Research Institute and School of Biological and Chemical Sciences, Queen Mary University of London, Mile End Road, London E1 4NS, UK. [5] Department of Advanced Materials Science, The University of Tokyo, 5-1-5 Kashiwanoha, Kashiwa, Chiba 277-8561, Japan. [6] JST, PRESTO, 4-1-8 Honcho, Kawaguchi, Saitama 332-0012, Japan. [7] Department of Applied Physics, Nagoya University, Chikusa, Nagoya 464-8603, Japan. [8] King Abdullah University of Science and Technology (KAUST), PSE, Thuwal 23955-6900, Saudi Arabia. [9] RIKEN Center for Emergent Matter Science, Wako, Saitama 351-0198, Japan. Correspondence and requests for materials should be addressed to H.S. (email: hs220@cam.ac.uk).

Organic semiconductors and conductors are enabling flexible, large-area optoelectronic devices, such as organic light-emitting diodes, transistors and solar cells. Due to their exceptionally long spin lifetimes, these carbon-based materials could also have an impact on spintronics, where carrier spins play a key role in transmitting, processing and storing information[1]. The recently observed inverse spin hall effect in organics[2,3], the conversion of a spin current to a transverse charge current, potentially enables fully organic spintronics devices.

Spin-orbit coupling (SOC), a relativistic effect which couples the charge's angular momentum to its spin, plays a dual role in such devices: it drives the spin to charge conversion and also provides a pathway for spin relaxation. Despite its fundamental importance for device applications, quantifying SOC strengths in organics based on investigating spin relaxation has proven difficult due to the competing and hard to distinguish contribution from hyperfine interactions (HFI)[4], the coupling of the charge's spin to nuclear spins. In fact, both SOC and HFI have been used separately to explain the magnetoresistance[5] and spin diffusion length[6] in spin valves of the same organic material.

The comparison of spin diffusion lengths $\lambda_S$ on the other hand has indeed revealed significant variations between different organic semiconductors[5,7,8] but separating effects of the charge carrier mobility and density[9] from the spin's coupling to its environment remains challenging. In addition, organic spin valve measurements can be affected by tunnelling magneto resistance through pin hole defects[10], which complicates an accurate determintion of $\lambda_S$.

A more direct probe of SOC is the voltage generated by spin to charge conversion (inverse spin hall effect voltage). However, such measurements often only provide the product of $\lambda_S$ and the spin Hall angle[3] and are prone to artefacts[11]. A systematic quantitative study of SOC strengths has therefore been hindered by the difficulty in isolating the effect of SOC on spin lifetimes, diffusion lengths or inverse spin hall voltages.

In this work we show that the g-factor (the isotropic part of a spin's coupling to an external magnetic field) of an unpaired spin from a charged molecule can be used as a measure of the effective SOC, over a wide range of SOC strengths. By effective SOC, we mean the overlap between the orbital- and spin angular momentum distributions, which respectively depend on the molecular composition and geometry, and the spin density in the charged molecule. Shifts of the g-factor from its free electron value ($\Delta g$) arise from SOC and orbital Zeeman terms in the Hamiltonian and are easily accessible by electron spin resonance (ESR). This provides a method to quickly and unambiguously determine the effective SOC over a wide range of light molecules without relying on indirect measurements. Our results demonstrate a remarkably rich variability—and therefore, potential for purposeful tuning—of molecular g-factors with subtle aspects of the molecular structure and composition.

In the second part of this communication, we correlate the above g-shifts to spin-lattice relaxation times determined by power saturation ESR measurements and demonstrate a change in spin lifetimes over four orders of magnitude, from 200 to 0.15 μs, with increasing $\Delta g$. This suggests that the change in g-factor can be indeed attributed to an increase in SOC and a corresponding reduction of the spin lattice relaxation time $T_1$.

While our simulations and measurements were all performed on isolated molecules, we argue that the gained information will be valuable to understand bulk systems, which in these systems are strongly influenced by the single molecule properties.

## Results

**Materials overview.** We compare a set of 32 fused-ring molecular semiconductors with systematically varying geometries and atomic substitutions, most of which are based on central thiophene or selenophene moieties that are commonly found in high-mobility organic semiconductors. Many of the chosen molecules such as 2,7-dioctyl[1]benzothieno[3,2–b][1]benzothiophene (C8-BTBT), 2,9-didecyl-dinaphtho[2,3-b:2′,3′-f]thieno[3,2–b]thiophene (C10-DNTT) or rubrene are widely studied and perform exceptionally well in organic thin film transistors with hole mobilities of $5–10 \, cm^2 V^{-1} s^{-1}$ and signs of coherent charge transport such as a metallic Hall effect[12] and a band-like temperature dependence of the mobility[13]. Since high charge carrier mobilites are expected to improve spin diffusion lengths[1] and the spin Hall angle is predicted to increase with reduced energetic disorder[14], such systems are natural candidates for spintronics applications. Hence, the chosen series of molecules offers a unique opportunity to systematically study the strength of SOC and its effect on spin lifetimes.

All ESR measurements were performed on radical cations in solution. To quantitatively understand these measurements we have performed density functional theory (DFT) calculations on positively charged molecules in the gas phase. This enables us to calculate spin density distributions with an accuracy that is not achievable for bulk systems and to experimentally resolve the HFI splitting of resonance lines. The latter acts as a probe of the spin density at nuclear coordinates throughout the molecule and provides an experimental method to validate the spin density calculations.

**Experimental g-shifts in organic molecules.** To achieve controllable experimental conditions with spin densities that can be reasonably compared to DFT calculations, we performed all ESR experiments on highly dilute solutions of radical cations in dichloromethane confined in capillary tubes at the cavity centre. The neutral molecules were oxidized in solution by adding aluminium chloride (AlCl$_3$) and the successful p-type doping was confirmed by optical spectroscopy (Fig. 1a,b)[15]. All ESR spectra were recorded on a Bruker E500 X-band spectraometer and as lock-in measurements with an external magnetic field modulation at 100 kHz. They therefore show the derivatives of the microwave absorption spectra. All experimental details are given in the Methods section.

ESR measures the resonant absorption of microwaves between Zeeman split energy levels in an external magnetic field **B**. The rapid motion of molecules in a solution causes an averaging over the anisotropic terms in the spin Hamiltonian so that the resulting resonances arise from

$$\mathcal{H}_{spin}^{iso} = g\mu_B \mathbf{S} \cdot \mathbf{B} + \sum_n a_n \mathbf{S} \cdot \mathbf{I}_n \qquad (1)$$

where the isotropic HFI couplings $a_n$ for the nuclear spin $\mathbf{I}_n$ are proportional to the spin density at the nth nucleus. As a result, we observe symmetric spectra where the isotropic g-factor $g = \hbar\omega/\mu_B B_0$ is directly given by the microwave frequency $\omega$ and the resonance centre $B_0$ ($2\pi\hbar$, Planck's constant). The isotropic HFI additionally causes a splitting of resonance lines proportional to $a_n$.

The experimentally investigated molecular structures and their measured and calculated isotropic g-shifts $g_{exp}$ and $g_{theo}$ are summarized in Table 1; the corresponding ESR spectra are shown in Supplementary Fig. 1. The series begins with the pure hydrocarbons rubrene and pentacene that are expected to show very weak SOC. The remaining set of molecules is based on the BTBT structure, which incorporates heavier sulfur atoms in two central thiophene units. For maximum consistency and rigour, theoretical calculations were performed on a set of systematic variations of this structure, including extending it with a single set of phenyl rings to form dinaphtho[2,3-b:2′,3′-f]thieno

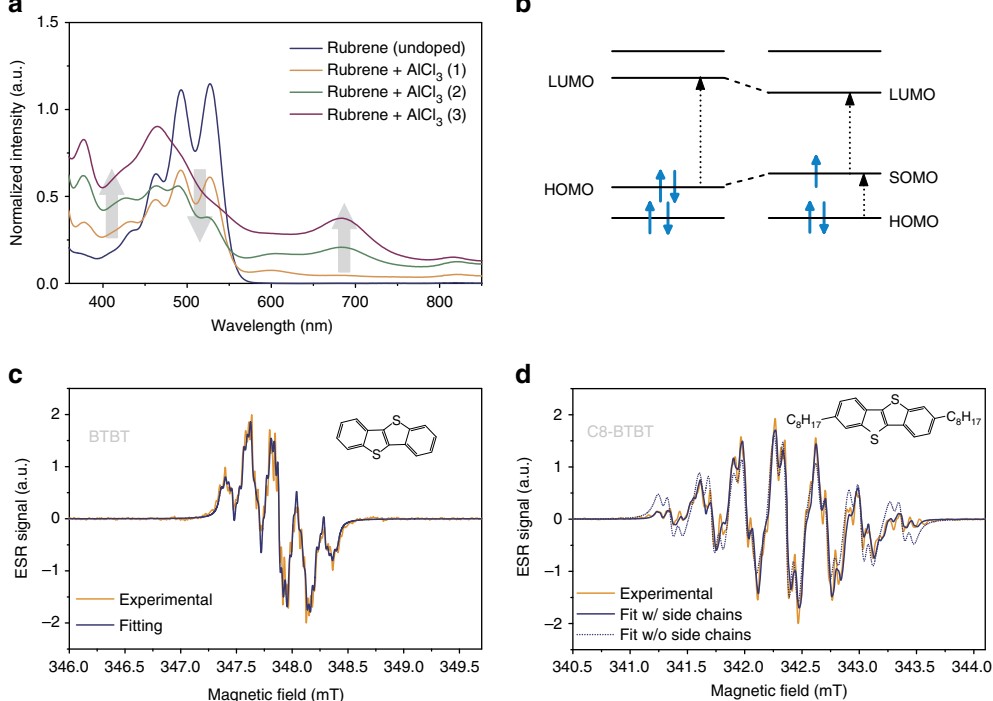

**Figure 1 | ESR and optical spectra of doped molecules.** (**a**) Optical absorption spectra of a $0.5 \times 10^{-3}$ mol l$^{-1}$ solution of rubrene in dichloromethane with increasing amounts of AlCl$_3$ (traces 1–3). We observe the expected bleaching of the transition between the highest occupied molecular orbital (HOMO) and the lowest unoccupied molecular orbital (LUMO) together with the emergence of charge induced transitions in the sub band gap regime from the HOMO to the singly occupied molecular orbital (SOMO). The grey arrows emphasize the increase of charge induced absorption and the decrease of the neutral molecule absorption peaks with higher doping levels. (**b**) A schematic of energy levels with possible optical transitions for a neutral and positively charged molecule. The shifts of the HOMO and LUMO energy levels reflects the reorganization of the molecular geometry to accommodate the charge. (**c,d**) Derivative ESR spectra of BTBT and C8-BTBT with best fits including (solid line) or excluding (dashed line) HFI from the first two hydrogens on the side chains.

[3,2–b]thiophene (DNTT), or two sets to form dinanthra [2,3-b:2′,3′-f]thieno[3,2–b]thiophene (DATT), adding a set of outer thiophene rings (dibenzothiopheno-thieno-thiophene, DTBTBT) in a curved (C-DTBTBT) or linear (L-DTBTBT) geometry and substituting the sulfur in all of these with selenium (BSBS, DNSS, DSBSBS and so on) for a total of 10 distinct geometries. Each of these structures was additionally examined with insulating alkyl chains attached to the outer phenyl or thiophene units (C8-BSBS, C10-DNTT and so on) and with the same side chains shifted one position closer to the closest central sulfur atom (labelled C8s-BTBT and so on, see Fig. 2a for the labelling of side chain positions), in total adding up to 32 distinct molecules. Supplementary Table 1 provides a complete list of all molecular names, structures and the corresponding acronyms.

From those 32 molecules, our experimental data is limited to the 11 molecules shown in Table 1 that were physically available. While those molecules have alkyl chain lengths between C8 and C12, all calculations were performed with C8 chains as explained in the Supplementary Note 1.

Some of the g-factor changes follow expected trends. As expected for holes in the highest occupied molecular orbital (HOMO) level with a strong coupling to lower lying $\sigma$-orbitals, all g-factors show positive shifts relative to the free electron value $g_e \approx 2.002319$. The light-element molecules pentacene and rubrene show g-factors close to $g_e$ and we see an increase of the g-shift by almost one order of magnitude for BTBT or DNTT with the inclusion of heavier sulfur atoms. Consistently, the next significant jump to $\sim 10^4$ p.p.m. takes place for BSBS and DNSS when replacing sulfur by selenium, reflecting the stronger SOC of selenium. On the other hand, there is only a minuscule

decrease in $\Delta g$, on the order of 100 p.p.m., with the addition of phenyl rings on either side of the molecule (BTBT→DNTT or BSBS→DNSS).

Other changes are more surprising. The addition of (insulating) alkyl chains to a molecule does not introduce or remove any atoms with strong SOC from the $\pi$-conjugated system, nor should it significantly alter the electronic structure of the latter[16]. Nevertheless, the g-shift is reduced by half with the introduction of side chains for BTBT→C8-BTBT and by about one quarter for DNTT→C10-DNTT.

To understand the origin of this reduction in $\Delta g$, we can exploit the isotropic HFI from the hydrogen nuclei as local probes of the spin density throughout the molecule. We determine the HFI coupling constants from the derivative ESR spectra by diagonalizing the spin Hamiltonian from equation 1 and convoluting the obtained resonance positions with a Lorentzian line to account for the finite line widths. The resulting resonance spectrum is then fitted to the experimental ESR spectrum, as shown in Fig. 1c.

When fitting the spectra of molecules with attached side chains, we had to include the HFI from the first two hydrogens on each side chain in order to reproduce the spectral shape (Fig. 1d for C8-BTBT and Supplementary Fig. 1). The number of hydrogens included in the fit provides clear experimental evidence for a spin density that leaks out onto the side chains and an accompanying change of the spin density distribution on the molecule.

Another striking experimental observation is the large difference in g-shifts between C-C12-DTBTBT and L-C12-DTBTBT. Both molecules are composed of exactly the same

**Table 1 | Simulated molecular *g*-shifts and experimentally determined values by ESR.**

| Acronym | Chemical name | Structure | $\Delta g_{exp}$ (p.p.m.) | $\Delta g_{theo}$ (p.p.m.) |
|---|---|---|---|---|
| rubrene | rubrene | | 309 | 372 |
| pentacene | pentacene | | 311* | 348 |
| BTBT | [1]Benzothieno[3,2–b][1]benzothiophene | | 2,141 | 2,008 |
| C8-BTBT | 2,7-Dioctyl[1]benzothieno[3,2–b][1]benzothiophene | | 1,087 | 809 |
| DNTT | Dinaphtho[2,3-b:2′,3′-f ]thieno[3,2–b]thiophene | | 1,959 | 1,990 |
| C10-DNTT | 2,9-Didecyldinaphtho[2,3-b:2′,3′-f]thieno[3,2–b]thiophene | | 1,657 | 1,646 |
| C-C12-DTBTBT | 2,8-Didodecyldibenzothiopheno[7,6-b:7′,6′-f]thieno[3,2–b]thiophene | | 354 | 420 |
| L-C12-DTBTBT | 2,8-Didodecyldibenzothiopheno[6,5-b:6′,5′-f]thieno[3,2–b]thiophene | | 3,514 | 4,180 |
| BSBS | [1]Benzoseleno[3,2–b][1]benzoselenophene | | 10,010 | 14,255 |
| C8-BSBS | 2,7-Dioctyl[1]benzoseleno[3,2–b][1]benzoselenophene | | 6,322 | 6,773 |
| DNSS | Dinaphtho[2,3-b:2′,3′-f ]seleno[3,2–b]selenophene | | 9,772 | 10,415 |

The uncertainty in experimentally determined *g*-factors is limited by the accuracy of the external magnetic field measurement of ±5 μT under standard conditions, resulting in an uncertainty of ±30 p.p.m. for the *g*-factors.
*Due to the extremely low solubility of Pentacene, the *g*-shift was measured for TIPS-Pentacene with a very similar conjugated system.

atoms and their only difference lies in the position of the outer thiophene units, either introducing a curve (C-C12-DTBTBT) or providing a more linear geometry (L-C12-DTBTBT). Even though both molecules contain four sulfur atoms, the *g*-shift changes from ∼4,000 to ∼400 p.p.m., when switching from the linear to the curved isomer. This arises from the difference in spin density distributions between the two molecules, as revealed by the changing HFI splitting in the ESR spectra (Supplementary Fig. 1), and is fully consistent with the molecular modelling discussed below.

The changes of the spin density distribution with alkylation, observed indirectly via the HFI, or the introduction of a curvature in the molecule suggest a strong dependance of the *g*-shift not only on the atomic composition of the molecules but also their geometries and the resulting spin densities. However, we cannot distinguish HFI couplings from different hydrogens experimentally unless they have different numbers of equivalent nuclei[17] and common sulfur or selenium isotopes have zero nuclear spins. DFT modelling on the other hand can provide an accurate, spatially resolved picture of the spin wave functions in the molecule.

**DFT modelling of *g*-shifts and spin densities**. To explain the experimental results above, and investigate the importance of the spin density distribution for $\Delta g$, we performed state-of-the-art DFT calculations of molecular *g*-tensors and spin densities. We have chosen an all-electron, hybrid exchange-correlation functional DFT level of theory, which also accounts for scalar relativistic and spin-orbit coupling effects. All computational details can be found in the Methods section and in Supplementary Note 1.

As evident from Table 1, the accuracy of the theoretically predicted $\Delta g$ is generally excellent, validating the quality of our methodology. The only significant discrepancy is found for BSBS as discussed in Supplementary Note 2 (Supplementary Fig. 3). In each of these calculations, the spin density in the cationic radical was calculated, plotted and visualized, as shown in Fig. 2a.

In the left of this figure, the qualitative effect of alkylation at the *outer* bonding site is shown using BSBS as an example. The effect is identical for the sulfur-based analogues, and similar but weaker in DNTT, DNSS, DATT and DASS. We attribute the reduced magnitude of the effect to the weaker charge confinement in these molecules.

In essence, the significant spin maximum (blue contour) at the *outer* bonding site causes the spin density to leak onto the alkyl chain, resulting in a net spin density depletion at the heavy atoms, visible in Fig. 2a as a diminished contour at the heavy atom site. Moving the alkyl chain to the shifted bonding site, which in the base molecule shows a spin density minimum (red contour) has a negligible or weakly opposite effect on the heavy atom spin density.

In the right of Fig. 2a, the effect of the geometry on the DTBTBT spin density is similarly shown, with the heavy atoms all but entirely depleted in the curved species. The same is observed in DSBSBS.

Comparing this visual analysis to the corresponding trends in *g*-shifts in Table 1 confirms the experimentally established picture of the spin density weight at heavy atoms, that is, the overlap of orbital and spin angular momentum distributions, as the key quantity for understanding *g*-shifts in this class of molecules. To verify this qualitative analysis, we proceed to build a quantitative model of the dependence of $\Delta g$ on the spin density. In the

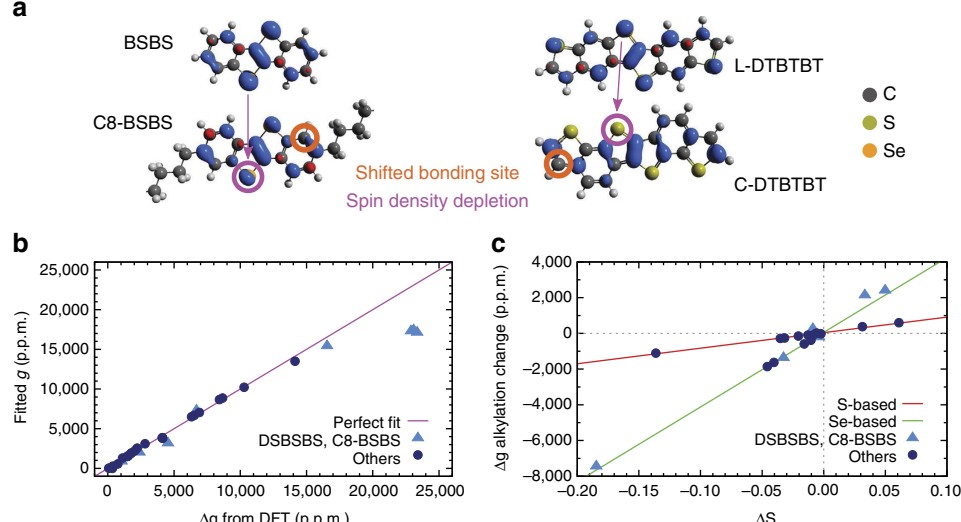

**Figure 2 | Relationship between g-shift and atomic spin densities.** (**a**) Spin isodensity contours of cationic (left) BSBS and C8-BSBS, and (right) L-DTBTBT and C-DTBTBT radicals. The C8-BSBS molecule shown has alkyl chains at the outer bonding site. Spin maxima and minima are shown in blue and red, respectively. The shifted bonding site at the phenyl rings has been labelled, and the observed spin depletion at heavy atoms highlighted. Note that only part of the alkyl chains in C8-BSBS are shown. (**b**) Correlation plot of $\Delta g^{OZ/SOC}$-terms calculated using DFT versus fitted on the form of equation 3. The outliers identified in the text are shown as light blue triangles, with the other numbers represented by dark blue circles. The magenta line $y = x$ represents a perfect fit. For brevity $\Delta g^{OZ/SOC}$ has here been relabelled $\Delta g$. (**c**) Plot of changes in $\Delta g^{OZ/SOC}$ terms as a function of change of effective heavy atom spin upon alkylation of molecules (see text). Red and green lines represent linear fits to the sulfur- and selenium-based molecules, respectively.

formulation of Neese[18], the molecular $\Delta g$ can be written as

$$\Delta g = \Delta g^{RMC} + \Delta g^{GC} + \Delta g^{OZ/SOC}, \qquad (2)$$

where $\Delta g^{RMC}$ is a relativistic mass correction, $\Delta g^{GC}$ is a diamagnetic gauge correction, and $\Delta g^{OZ/SOC}$ is a cross-term between the orbital Zeeman and spin-orbit coupling operators corresponding to the mixed derivative of the molecular total energy with respect to the electron magnetic moment and external magnetic field[18]. The first of these terms, like other relativistic effects, is generally negligibly small in light organic molecules. The second term is in the current set of molecules also comparably small and of opposite sign, nearly cancelling the first term. g-shifts in this set are therefore virtually identical to the third term, $\Delta g^{OZ/SOC}$.

To establish a quantitative model that relates the g-shift to the spin density distribution, we use an Ansatz with a linear dependence of $\Delta g^{OZ/SOC}$ on the local spin density at each atom:

$$\Delta g^{OZ/SOC} \approx \sum_{e=1}^{N} c_e \sum_{n=1}^{N_e} \sigma_n^e, \qquad (3)$$

Here, $\sigma_n^e$ is the effective spin at atom $n$ of element $e$, $N_e$ is the number of atoms of that element in the molecule, $N$ is the number of different elements and $c_e$ is a proportionality constant of element $e$. The effective atomic spin $\sigma_n^e$ is here treated in units of electronic spin (that is, on the scale of $\frac{\hbar}{2} = 1$), and therefore as dimensionless. The proportionality constants represent the net effect of the orbital angular momentum operators in $\Delta g^{OZ/SOC}$. This model then describes g-shifts as dependent on the overlap of orbital and spin angular momentum distributions, or equivalently, an effective spin-orbit coupling. We prove its validity by studying the correlation between g-shifts calculated from DFT, and shifts fitted to the same on the form of equation 3.

The results of this fit are shown in a correlation plot in Fig. 2 and the fit coefficients $c_e$ are given in Supplementary Table 2. Using the coefficient of determination, or '$R^2$-value' as a quality measure of the fit, we note that a straight fit of atomic spins to calculated $\Delta g^{OZ/SOC}$ values of the full 32 molecule set yields a

fairly low $R^2$ of 0.960. For a strictly linear statistical hypothesis, this is unconvincing. However, the statistic is straightforwardly divided into (a) eight outliers consisting of all the Se-substituted DTBTBT molecules (that is, {C-,L-}{C8-,C8s-}DSBSBS), and the alkyl-functionalized, Se-substituted BTBT molecules ({C8-,C8s-}BSBS), and (b) the 24 molecules in the rest of the set. The outliers and the rest are shown in Fig. 2b as light blue triangles and darker blue circles, respectively. Excluding the outliers from the fit yields a very high $R^2$ of 0.996, showing that the model works well for the pure hydrocarbons, all the sulfur-based molecules, the non-functionalized BSBS molecule, and all variations of the DNSS and DASS molecules.

The reason why the Ansatz of equation 3 works better for sulfur than selenium is the non-local terms scaling with the smaller orbital angular momentum in the former. Furthermore, the dual heavy-atom structure in the DSBSBS molecules significantly complicates their electronic structure, with much larger non-local terms compared to the other, simpler molecules. The alkyl-functionalized BSBS molecules are also included in the outliers because the positive charge is much more strongly confined in BSBS than in the larger DNSS and DASS. As will be elaborated below, the spin density of BSBS is consequentially more strongly perturbed when chains are added, leading to stronger non-local interactions and violating the premises of the linear model.

With the range of applicability of this model thus established, we continue by applying it to studying the effect of alkyl-chain functionalization. In Fig. 2c, the difference in $\Delta g^{OZ/SOC}$ upon alkylation has been plotted, as a function of the corresponding difference $\Delta S$ in effective spin at the heavy atoms of the molecule. For clarity, the sets of outliers and the rest in Fig. 2b have been coloured the same way.

Two things stand out in Fig. 2c. First, the relationship between the change in the spin density on the sulfur-atoms and the change in g-shift on alkylation is linear. This fact explains the, in parts, dramatic alkylation effects on g-shifts. Such large shifts occur when the alkyl chain bonds to a site where the spin density is large in the non-functionalized molecule. The spin density will

then spread partly into the chain and will be reduced on the heavy atom, hence allowing for tuning of the effective spin-orbit coupling by targeted alkylation.

Second, the outliers of Fig. 2b are now a much better fit in Fig. 2c. This is due to some non-local terms cancelling when differences are taken between alkylated and non-alkylated g-shifts. Of the outliers, C8-BSBS and C8 s-BSBS respectively exhibit the strongest decrease and increase of spin at the heavy atoms, and are found in Fig. 2c at either extreme of the Se-based statistic. As explained above, this strong alkylation effect is attributed to the confinement of the electron hole in the small BSBS molecule leading to a large shift of the spin density distribution. This point is additionally reinforced by the Se-based molecules not among the outliers: For C8-DNSS and C8 s-DNSS, the electrons are less confined to begin with and the effect of attaching an alkyl chain is therefore less pronounced.

**Relationship between g-shift and spin lifetimes**. In the previous section, we have argued that the g-shift can be correlated to an effective SOC over a wide range of g-factors. This series of molecules therefore presents a unique opportunity to systematically study the effect of SOC on spin lifetimes. We show in the following that the spin lattice relaxation time $T_1$ indeed scales with $\Delta g$ over four orders of magnitude and interpret this in the framework of the Bloch-Wagness-Redfield theory of spin relaxation.

The majority of reported experimental architectures in (organic) spintronics employ an external magnetic field **B** to either generate or control and manipulate spins inside the target material. The employed fields of 10–400 mT (refs 19,20) are on the same order or smaller than in ESR measurements. In the presence of an external field one must distinguish between longitudinal and transverse spin relaxation. The first manifests itself macroscopically as the decay of the sample magnetization parallel to **B** and requires transitions between the Zeeman split spin-up and spin-down states. This implies an exchange of energy with the spin's environment (the 'lattice') and is characterized by the spin lattice relaxation time $T_1$. The decay of the transverse magnetization is characterized by the coherence time $T_2$. It results from spin dephasing and can occur in an energetically neutral process. A long $T_2$ enables the coherent manipulation of spins while a large $T_1$ is crucial to maintain a spin polarization along a preferred axis.

We determine $T_1$ and $T_2$ from the power saturation behaviour of the ESR resonances (Fig. 3) and the unsaturated line width, respectively, as described in Supplementary Note 3. This requires both a sufficiently large microwave field $B_1$ and an exact knowledge of the latter over the sample volume. We therefore minimize the variation of $B_1$ by confining the sample to a capillary tube at the cavity centre and account for the lateral field distribution across the sample, as shown in Fig. 3 and Supplementary Fig. 3.

The resulting values for $T_1$ reveal a strong correlation between spin lattice relaxation times and g-shifts: we observe a change in $T_1$ over four orders of magnitude, from 212 μs for molecules with small g-shifts (for example, C-C12-DTBTBT) down to 0.15 μs (BSBS, DNSS) for the largest g-shifts (Fig. 4a). This demonstrates the impact of tuning the g-factor beyond changing the coupling of a spin to the external magnetic field. By increasing the spin density at selenium or sulfur atoms, we increase the effective SOC for the spin and as result the transition rate between spin-up and spin-down levels increases as well.

The spin coherence times, $T_2$, are systematically smaller than $T_1$ and show a less pronounced dependence on $\Delta g$ and a larger scatter of values. This can be understood when considering the underlying mechanism that contribute to $T_2$. A spin couples to its

environment via magnetic fields and the exchange of energy required for spin lattice relaxation implies that the transitions between Zeeman levels are induced by fluctuating fields. Those typically are internal HFI and SOC fields which fluctuate with lattice vibrations, the tumbling motion of molecules in a solution or (in the rest frame of the spin) due to the spatial motion of the charge carrier in a solid state system. In the framework of the Bloch–Wagness–Redfield theory[21,22], those fields are treated classically by adding a time dependent perturbation $\mathcal{H}_{\mathrm{pert}}(t) = \mathbf{S} \cdot \mathbf{F}(t)$ to the spin Hamiltonian and examining the spectral densities $k_{x,y,z}(\omega)$ of the fluctuating fields $F_{x,y,z}(t)$. The spin lattice relaxation time then depends on the spectral density at the Larmor frequency $\omega_L = \gamma_e B$ while the spin coherence time includes an additional contribution from $k(0)$[22]:

$$T_1^{-1} = \frac{1}{\hbar^2} \left[ k_x(\omega_L) + k_y(\omega_L) \right] \tag{4}$$

$$T_2^{-1} = (2T_1)^{-1} + \frac{1}{\hbar^2} k_z(0) \tag{5}$$

Equations 4 and 5 imply that $T_2 \lesssim T_1$, which is consistent with our measured values. We also see that $T_2$ is susceptible to static local variations such as small inhomogeneities in the external magnetic field. This effectively creates an upper bound for $T_2$ or equivalently a lower bound for the line width ($\sim 3 \times 10^{-3}$ mT) and introduces larger fluctuations between measurements, for example, due to differences in the sample position, that do not affect $T_1$.

If we consider relaxation caused by SOC fields only, the fluctuations can be written as $\mathbf{F}(t) = \mu_B \Delta \mathbf{g}(t) \cdot \mathbf{B}$. This incorporates a time dependance from lattice vibrations or molecular tumbling. Assuming that the amplitude of the SOC fluctuations approximately scales with $\Delta g$ itself, one expects that the spectral densities follow the proportionality $k_q(\omega) \propto (\Delta g)^2$ and the relaxation time to follow $T_1 \propto (\Delta g)^{-2}$.

Figure 4a shows that $T_1$ indeed follows this proportionality with a remarkable accuracy. We conclude that spin lattice relaxation is therefore dominated by the effective SOC at magnetic fields of $\sim 350$ mT.

Note that fluctuating HFI fields create perturbations of the form $\mathbf{F}_n(t) = \mathbf{A}_n(t) \cdot \mathbf{I}_n$, where $\mathbf{A}_n(t)$ is the coupling tensor for the nuclear spin $\mathbf{I}_n$. This expression does not depend on the external field and the HFI therefore become less effective at flipping the spin when the Zeeman splitting increases at higher fields. In contrast, spin lattice relaxation by fluctuating SOC fields will remain equally effective at all external field strengths. Measurements on organic spin valves for instance are typically performed at different magnetic fields between 5 and 500 mT, depending on the switching field of the ferromagnetic electrodes. The effect of HFI fields on spin relaxation will therefore be suppressed to different degrees which complicates a systematic comparison of spin diffusion lengths.

Only two outliers, TIPS-pentance and rubrene, stand out for their small $T_1$ relative to the g-shift. Both have a large number of hydrogens attached to the conjugated system and we speculate that the HFI provides an additional relaxation pathway. Naturally, this would be most pronounced for rubrene, where 28 protons contribute to the HFI. In a control experiment, we repeated the measurement on fully deuterated d28-rubrene, which strongly suppresses hyperfine interactions[8,23]. Together with a reduction of the HFI couplings by a factor of 2–3, we observe an increase of $T_1$ by more than an order of magnitude which brings it in line with the other molecules.

The large deviation from the $T_1 \propto (\Delta g)^{-2}$ dependency can therefore be traced back to hyperfine interactions. We can quantify their contribution by summing over SOC and HFI

relaxation rates $T_1^{-1} = \left(T_{1,\mathrm{HFI}}\right)^{-1} + \left(T_{1,\mathrm{SOC}}\right)^{-1}$ which yields pure HFI relaxation times of $\sim 13\,\mu s$ for h28-rubrene and $\sim 30\,\mu s$ for TIPS-pentacene. Such an estimate is only meaningful when the deviation from $T_1 \propto (\Delta g)^{-2}$ is larger than the measurement uncertainty, that is, when the contribution of $T_{1,\mathrm{HFI}}$ to $T_1$ is not negligible. We therefore cannot systematically extract HFI contributions for most of the molecules in this study but instead can identify a domain of weak SOC with $\Delta g \lesssim 500$ p.p.m., where, in the presence of a sufficient number of nuclear spins, hyperfine fields will significantly contribute to spin lattice relaxation.

When excluding the outliers and assuming that both $T_1$ and $T_2$ are dominated by a single relaxation process, SOC in our case, one can find a simple expression for the spectral density $k_q(\omega) = \overline{F_q^2}\tau_C/(1 + \omega\tau_C^2)$, where $\tau_C$ is the correlation time of field fluctuations and $\overline{F_q^2}$ is the mean square of the fluctuating field[22]. Figure 4c shows the dependence of both relaxation times on $\tau_C$ in this model and demonstrates that spin lattice relaxation becomes most effective when the frequency $\tau_C^{-1}$ matches the energy difference between the Zeeman split levels.

In a rough approximation, we can take $\overline{F_x^2} = \overline{F_y^2} = \overline{F_z^2}$ and estimate the correlation time $\tau_C$ from the ratio $T_1/T_2 = 1 + \omega_L^2\tau_C^2/2$. The resulting values are on the order of $\tau_C^{-1} = 2.5 - 50\,\mathrm{GHz}$ (Fig. 4c). For comparison, simulations of intramolecular vibrational modes of DNTT predict frequencies of 6–50 THz for stretching modes along the molecule's symmetry axes, distortions of the phenyl rings and vibrations of the C-H bonds, in ascending order[24]. They exceed the Zeeman splitting by orders of magnitude and are therefore less effective at flipping the spin.

Alternatively, we consider SOC fields that are static in the molecule's rest frame but fluctuate relative to the external field (and thus the spin quantization axis) with the tumbling motion of molecules in a solution. From diffusion-ordered nuclear magnetic resonance spectroscopy (DOSY NMR) measurements at a concentration of $0.2\,\mathrm{mg\,ml}^{-1}$, we can calculate the Stokes radius and resulting rotational correlation times for the molecules (Supplementary Note 4). The solubility of DNTT and DNSS in dichloromethane was too small to record DOSY NMR spectra but the rotational correlation times $\tau_C^{\mathrm{diff}}$ for the remaining molecules are shown in Fig. 4c together with $\tau_C$ values estimated from the spin lifetime ratios. Even though the values of $\tau_C$ are only rough estimates, derived under the previously discussed assumptions, we observe an approximate agreement with $\tau_C^{\mathrm{diff}}$. It is therefore likely that the observed relaxation times are indeed a result of SOC fields which fluctuate due to the molecular rotations in solution.

## Discussion

Isolated molecules, both in the gas phase and in solution, are significantly easier to model and to understand than solid thin films. They also provide us with the unique opportunity to resolve the hyperfine structure of ESR signals and to verify the spin densities from relativistic DFT modelling by comparing experimental and theoretical HFI couplings. Altogether with accurate measurements and predictions of the g-factor, we demonstrate the subtle dependence of SOC strengths on the molecular geometry: the inclusion of heavier atoms in the molecular structure increases SOC but the effect of such substitutions can be almost completely suppressed by the molecular geometry and the resulting spin density distribution. As a result, molecules such as C-C12-DTBTBT with four sulfur atoms exhibit SOC strengths close to pure hydrocarbons. We demonstrate an empirical linear relationship between the g-shift and the atomic spin densities, where the proportionality constants depend only on the type of element and its local SOC, regardless of its molecular

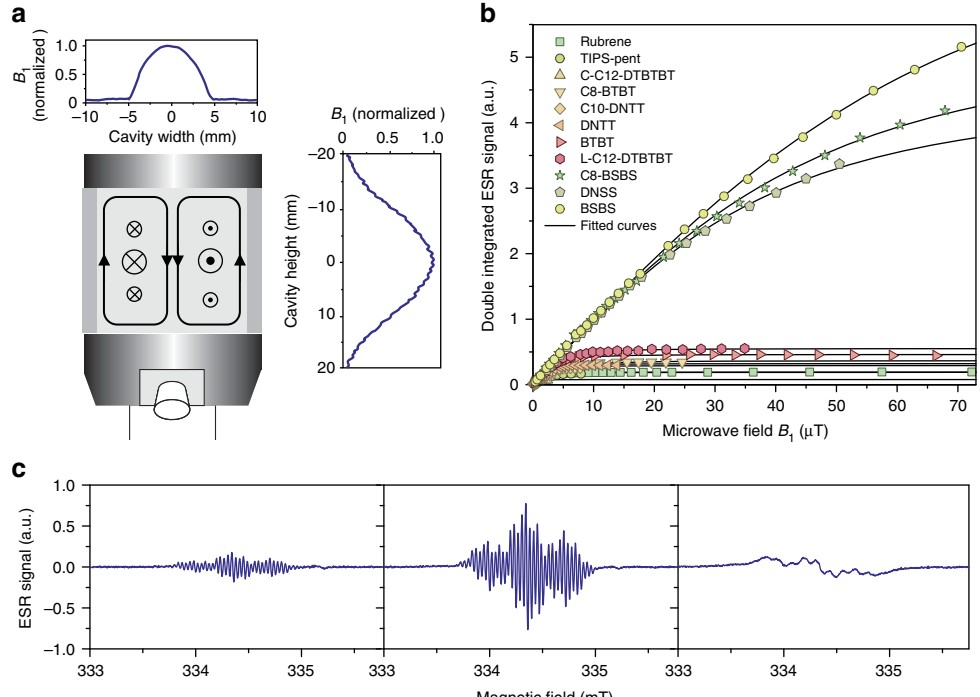

**Figure 3 | Power-saturation measurement of spin lifetimes.** (**a**) Schematic of the Bruker ER 4122SHQE cavity with microwave magnetic and electric fields in-plane and out-of-plane, respectively. Plot insets show the magnetic field strength across the cavity. We account for the vertical distribution of $B_1$ and minimize horizontal variations by confining our samples to a diameter of 1 mm. (**b**) Power saturation of the integrated ESR absorption spectra for the series of molecules with fitted curves to extract $T_1$ and $T_2$. (**c**) Evolution of derivative ESR spectrum with increasing microwave powers for the example of L-C12-DTBTBT.

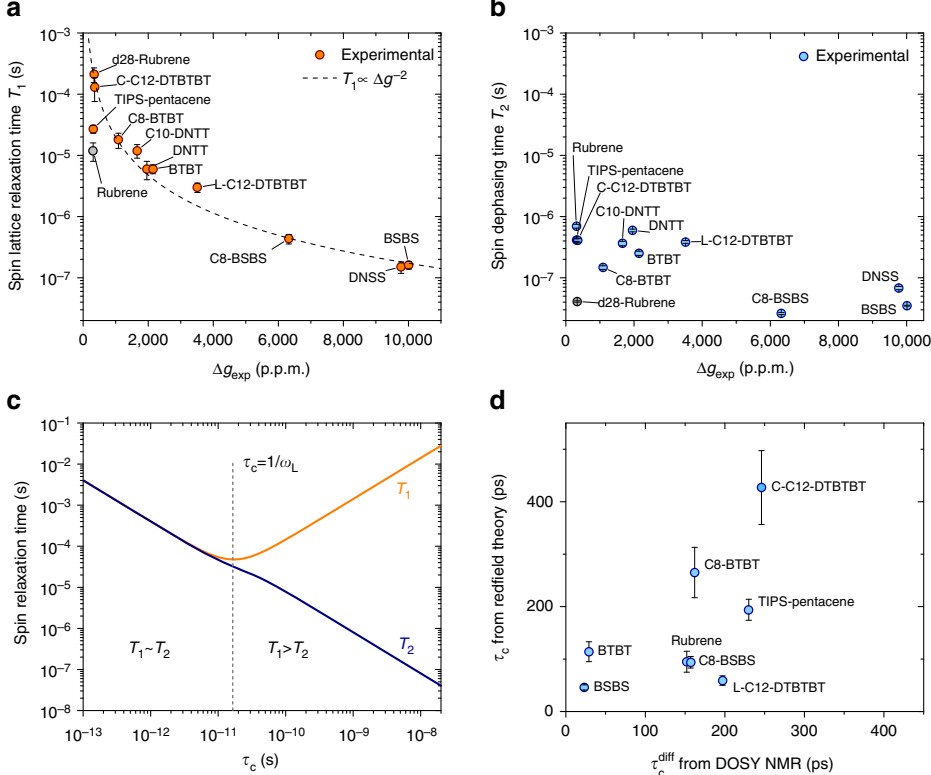

**Figure 4 | Dependance of spin relaxation times on the effective SOC. (a)** Spin lattice relaxation time versus $\Delta g$ for all measured molecules. Error bars represent 95% confidence intervals. Dashed line shows the expected proportionality $T_1 \propto (\Delta g)^{-2}$ for relaxation via SOC fields. **(b)** Spin coherence time versus $\Delta g$ for all measured molecules. Error bars show the 95% confidence intervals. **(c)** Dependence of $T_1$ and $T_2$ on the correlation time $\tau_C$ of field fluctuations from the Redfield theory. Model values of $\omega_L/2\pi = 9.4$ GHz and $B^2_{x,y,z} = 0.2$ mT were used for the plot. **(d)** Correlation times $\tau_C$ as estimated from the Redfield theory and plotted against rotational correlation times obtained from DOSY NMR diffusion constants (when available). Error bars from 95% confidence intervals of $T_1$ and $T_2$ and error propagation.

environment. This enables the targeted molecular design of materials with excellent charge transport properties and suppressed or enhanced spin-orbit coupling for either spin transport or spin to charge conversion applications.

The mechanism of spin relaxation in solution, which is most likely dominated by molecular tumbling, cannot be directly transferred to solid systems. Nevertheless, we expect to see some parallels. First ESR measurements on field induced charges in thin films reveal relaxation times that come close to the solution values and agree with pulsed ESR data from thin films at room temperature[25,26]. In the solid state, the fluctuations of SOC fields will not be provided by molecular rotations but by intermolecular phonon modes which lie at smaller frequencies than their intramolecular counterparts or by the hopping motion of spins in the organic semiconductor. When a charge carrier hops between sites with differently oriented molecules, it will experience the change in g-tensor orientation as a fluctuating field in its rest frame. Hopping times are estimated to be in the range of 100–1,000 ps (ref. 27) close to the rotational correlation times in solution and some lower frequency intermolecular modes can reach time scales > 10 ps (refs 28,29). In the solid state, the spin densities are likely to spread over multiple molecules but we expect $\Delta g$ to still be measure of the effective SOC. Since the fluctuation amplitudes will still scale with the latter, knowing the relaxation times in solution should provide a good estimate for both the coherence and spin lattice relaxation times in thin films.

## Methods

**ESR experiments.** All ESR spectra were recorded on a Bruker E500 X-band spectrometer with a Bruker ER 4122SHQE cavity at frequencies of 9.4–9.7 GHz and

with a concentration of cations below $0.5 \times 10^{-3}$ mol l$^{-1}$ to prevent interactions between spins on different molecules. The spectra were recorded at low-microwave powers of 0.006–0.06 mW to prevent power saturation and the concurrent line width broadening (Fig. 3c). The external magnetic field was modulated at 100 kHz and the spectra were recorded in lock-in measurements. As a consequence, passage effects can distort the signal shape if the spin packets cannot relax between modulation cycles. This was avoided by choosing a sufficiently small modulation amplitude $B_m$ or frequency $\omega_m$ so that $\omega_m B_m \ll 1/(\gamma_e T_1 T_2)$ and the spin system remains continually in thermal equilibrium[30].

The solutions of molecules in dichloromethane were prepared in a nitrogen glove box and sealed in sample tubes before the measurement. To minimize the perturbation of the microwave field inside the cavity, the solution was confined to a capillary tube with an inner diameter of 1 mm at the cavity centre where the magnetic field is largest (Fig. 3a). This allows us to perform the measurements at high-quality factors of $Q \approx 8,000$–10,000 (the ratio between the power stored and the power dissipated per microwave cycle).

The molecules were doped by adding AlCl$_3$, an oxidative Lewis acid, and most solutions remained stable for > 12 h (confirmed by both ESR and optical spectroscopy). Acquiring a full series of spectra with varying microwave powers took at most 1.5 h. After 24 h we could observe a noticeable reduction in the number of cations but no spectral changes that would indicate a reaction of AlCl$_3$ with the molecules. The only exception to this are C8-BTBT and C8-BSBS, which exhibit the emergence of a secondary spectrum with a smaller g-factor but partially overlapping with the organic radical signal. The new species are likely to be chlorinated versions of the molecules. We therefore conducted all C8-BTBT and C8-BSBS measurements immediately after adding AlCl$_3$ and before any spectral changes took place.

The fitting of all spectra was performed with the genetic algorithm of the EasySpin Toolbox[17] in MATLAB with a population of 40 spectra and 20,000 generations. Each fitting step was performed by diagonalizing the above Hamiltonian for a set of hyperfine couplings $a_n$ and determining allowed microwave transitions between the resulting energy levels. The resonance positions were then convoluted with a Lorentzian line to account for the finite spin dephasing time with $\Delta B_{1/2}$ as an additional fitting parameter.

**DFT modelling.** All density-functional theory calculations were performed using the NWChem[31] (version 6.5) and ORCA[32] (version 3.0.3) quantum chemistry

software packages using an all-electron DFT method, with nuclear relativistic effects described by the zeroth-order regular approximation (ZORA[33]) using the standard point-charge approximation for the atomic nuclei. g-tensors were calculated using the method[18] developed by Neese et al.[34] and related techniques as implemented in the ORCA software package. Atom spin populations were calculated using the Voronoi deformation density method[35], as implemented in the 'bader' program developed by Henkelman et al.[36], version 0.95a.

The fit of equation 3 is performed as follows: for each molecule, the atomic spin populations are summed up per element. Then, the proportionality constants per element $c_e$ of equation 3 are fitted using all other molecules in the fit statistic. The molecule fitted for is thus never part of the fit statistic. Finally, the fitted value of $\Delta g^{OZ/SOC}$ is calculated as the scalar product between the arrays of per element atomic spin population sums and the corresponding fitted proportionality constants $c_e$. As a negative $c_e$ lacks physical meaning in this model, a non-negative multivariate linear regression method (subroutine 'scipy.optimize.nnls'), as implemented in the Scientific Python software package, version 0.16.1, was used. As an approximate statistical test, we for each fit calculate the $R^2$-value according to

$$R^2 \equiv 1 - \frac{SS_{res}}{SS_{tot}} = 1 - \frac{\sum_i (x_i - y_i)^2}{\sum_i (x_i - \bar{x})^2} \qquad (6)$$

where $SS_{res}$, $SS_{tot}$, $x_i$ and $y_i$ are the residual sum of squares, the total sum of squares, and the calculated and fitted $\Delta g^{OZ/SOC}$ value, respectively.

**Calculation of spin lifetimes.** For sufficiently large microwave powers, spin relaxation will be too slow to maintain a population difference between the Zeeman split energy levels. The absorbed microwave power, integrated over the whole spectrum, will saturate. The integrated area of the ESR absorption spectrum, or the double integral over the derivative spectrum, follows[37,38]

$$DI(B_1) \propto \frac{B_1}{\sqrt{1 + \gamma_e^2 T_1 T_2 B_1^2}}. \qquad (7)$$

Fitting the power saturation curves of Fig. 3b therefore allows us to determine the product of spin lifetimes $T_1 T_2$.

Before the onset of power saturation and line width broadening, the coherence time is given by $T_2 = (\gamma_e \Delta B_{1/2})^{-1}$, where $\Delta B_{1/2}$ is the half width at half maximum of the individual Lorentzian resonance lines that make up the spectrum[37]. We determine $\Delta B_{1/2}$ by the least-squares fitting of the hyperfine structure and dividing the product of spin lifetimes by $T_2$ then gives the spin lattice relaxation time $T_1$.

**Data availability.** The data that support the findings of this study are available at the University of Cambridge data repository at https://doi.org/10.17863/CAM.8126.

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

## Acknowledgements

S.S. thanks the Winton Programme for the Physics of Sustainability, the Engineering and Physical Sciences Research Council (EPSRC), C. Daniel Frisbie for supplying d28-rubrene and Shin-ichi Kuroda for useful discussions. Funding from the Alexander von Humboldt Foundation, ERC Synergy Grant SC2 (No. 610115), and the Transregional Collaborative Research Center (SFB/TRR) 173 SPIN + X is acknowledged.

**Author contributions**

S.S. carried out the ESR experiments and analyses, E.R.M. performed the simulations and analyses. C.B.N. performed the DOSY NMR measurements. H.-Y.C., K.T. and I.M. provided materials. H.S. and J.S. supervised the project. All authors discussed the results. S.S. and E.R.M. wrote the manuscript with input from all authors.

**Additional information**

**Competing interests:** The authors declare no competing financial interests.

