## [Peer Review File · Nature Communications]

Reviewers' comments:

Reviewer #2 (Remarks to the Author):

The authors studied the g-shifts of isolated molecules using an EPR apparatus and found the relation between SOC strength and molecular composition and structure in a series of high mobility molecular semiconductors. The spin-lattice relaxation times vary significantly with the g-shifts by several orders of magnitudes. Although I found the study interesting, but there are several key questions that need to be addressed:

- 1) The authors did not interpret why the SOC strength strongly depends on the structure rather than its composition in some cases? What is the underlying mechanism for this?
- 2) The spin-lattice relaxation times in their result varies largely in the molecules. However, the studies of organic spin valves showed that the spin diffusion length is very similar for most studied materials. Why? Is it because of the isolation of the molecules in solution? What is the contribution of the HFI in addition to the SOC on the relaxation times in their result?
- 3) A similar topic has been studied by Drew et al. (PRL 110, 216602 (2013)) using a low energy muon spin rotation apparatus. The SOC strength was studied in solid state films rather than solutions. This is supposed to be more realistic than the study of SOC in solution. They found that the SOC strength does not depend on the molecular structures but the composition in the chemical backbone. This is in contradiction to the result obtained by the authors. I think due to the lack of the inter-molecular interaction, the isolated molecules in solution might have very different spin interaction in comparison to that in the solid state films. How would the authors design experiments to rule out this case?
- 4) One of the most important issues in organic spintronics in these days is how to distinguish the SOC and HFI effect. The authors need to extract the HFI result from their data for the comparison.

Reviewer #3 (Remarks to the Author):

This manuscript reports a study into the tuning of spin-orbit interaction (SOI) in organic semiconductors. This is an area of wide interest, as pointed out by the authors, and where there has been significant experimental interest over the last 5-10 years. It should be stressed that much of the most frequently cited literature on spin interactions in organic semiconductor, particularly with respect to organic spin valves and organic magnetoconductance, has been relatively empirical and to an extent polarised with a significant community who have claimed the SOI are less important in organic semiconductors (due to the low mass of most of the constituent atoms). This manuscript provides a significant development in that it not only shows that not only can SOI be an important spin interaction mechanism but also highlights the circumstances under which hyperfine interactions can dominate. As such the work is a highly valuable contribution and helps to move the research field into a more balanced regime where the effect of the different contributions can be clearly assessed. The result of this is that the authors have been able to determine rules by which the effective spin-orbit coupling can be engineered through molecular design in order to achieve the desired properties for a range of spin based devices.

I strongly believe that this manuscript marks a significant development in our ability to control spin interactions in organic semiconductors and hence that it should be published in nature Communications.

I do have a few relatively minor changes that I would like to see in order to improve the readability of the work for a broad audience. These are primarily on the definition of acronyms within the manuscript. As is common with many works on organic semiconductors there are a wide range of

acronyms that are used for the molecules in order to make the work readable. With 32 different molecules this becomes an absolute necessity. However, not all of the acronyms are defined (e.g. DNTT and DATT on page 3, BSBS, DNAA on page 4). As it may make the reading of the manuscript difficult to follow if they are each spelt out in the text I would suggest that an expanded version of table 1 and S2 (to include all molecules, acronyms, molecular structure and chemical name) should be included in the supplementary information. This would particularly benefit the non-chemists reading the manuscript. On a similar note the use of $D\{N,A\}\{TT,SS\}$ is "obvious" only after I've gone back through the whole manuscript with the sole purpose of trying to make sense of it. Similarly although SOMO is well known by those in the field (Fig 1(a)) it should still be defined.

Reviewer #2 (Remarks to the Author):

The authors studied the g-shifts of isolated molecules using an EPR apparatus and found the relation between SOC strength and molecular composition and structure in a series of high mobility molecular semiconductors. The spin-lattice relaxation times vary significantly with the g-shifts by several orders of magnitudes. Although I found the study interesting, but there are several key questions that need to be addressed:

We thank the reviewer for the generally positive assessment of our work and acknowledge that there are still some questions that require clarification. In the following, we aim to address those concerns.

1) The authors did not interpret why the SOC strength strongly depends on the structure rather than its composition in some cases? What is the underlying mechanism for this?

The dependence of the SOC strength on structure as well as composition is a key result of our work, and one we feel we devote much effort to interpreting. The fifth paragraph of the introduction (lines 56 – 65) has been rewritten to further clarify this interpretation, which we hope will meet with the approval of the reviewer:

In this work we show that the g-factor of a charged molecule can be used as a measure of the effective SOC, over a wide range of SOC strengths. By effective SOC, we mean the overlap between the orbital- and spin angular momentum distributions, which respectively depend on the molecular composition, geometry and the spin density in the charged molecule. Our results demonstrate a remarkably rich variability - and therefore, potential for purposeful tuning - of molecular g-factors with subtle aspects of the molecular structure and composition. Shifts of the g-factor from its free electron value (Δg) arise from SOC and orbital Zeeman terms in the Hamiltonian and are easily accessible by electron spin resonance (ESR). This provides a method to quickly and unambiguously determine the effective SOC over a wide range of light molecules without relying on indirect measurements.

2) The spin-lattice relaxation times in their result varies largely in the molecules. However, the studies of organic spin valves showed that the spin diffusion length is very similar for most studied materials. Why? Is it because of the isolation of the molecules in solution? What is the contribution of the HFI in addition to the SOC on the relaxation times in their result?

We agree these are valid general questions, but we believe that a discussion of spin diffusion lengths in organic spin valves is beyond the scope of our work. In fact, any attempt to interpret spin diffusion lengths reported for different organic semiconductors from spin valve measurements would be riddled with difficulties as in our opinion a significant fraction of organic spin valve papers in the literature is potentially affected by experimental artefacts and other complications: Vertical spin valves exhibit strong interface effects and possible tunneling magneto resistance through pin hole defects as discussed by Göckeritz et al., APL 106, 102403, 2015. This is in fact one of the reasons, why the focus of our paper is on the quantification and manipulation of spin-orbit coupling and its effect on spin life times using a simpler (ESR) method that allows a more reliable, direct comparison between organic materials than would be possible for spin valves, in which the quality of thin films necessarily varies between different organic materials. Our study has of course implications for spin-to-charge conversion, intersystem crossing in organic LEDs and of course organic spin valves.

Furthermore, it may also not be correct that “the spin diffusion length is very similar for most studied materials”. Reported spin diffusion lengths (λ_S) do change over orders of magnitude for different organic materials:

Material	λ_{SDL}	Source
DOO-PPV	16 nm	Nguyen et al., Nat. Mat. 9 , 345–352, 2010
C ₆₀	150 nm	Zhang et al., Nat. Comm. 4 , 1392, 2013
PBTTT	200 nm	Watanabe et al., Nat. Phy. 10 , 308, 2014
Alq3	45 nm	Z. H. Xiong et al. , Nature, 427 , 821, 2004

In addition, the spin lifetime is only one parameter governing λ_S which also strongly depends on the out-of-plane mobility μ via the Einstein relationship. The latter varies significantly between different organic materials and complicates a direct comparison between λ_S and T_1 .

We have added a paragraph to the introduction (lines 46-50) which motivates our decision to investigate g-shifts and spin lifetimes instead of spin diffusion lengths:

The comparison of spin diffusion lengths λ_S between different organic semiconductors has indeed revealed significant variations^{5,7,8} but separating effects of the charge carrier mobility and density⁹ from the spin's coupling to its environment remains challenging. In addition, organic spin valve measurements can be affected by tunneling magneto-resistance through pin hole defects¹⁰ which complicates the extraction of accurate spin diffusion lengths.

Due to the above complications, we prefer not to speculate about λ_S in our set of materials. We do not perceive any inconsistencies between our spin lifetime measurements and published spin diffusion lengths.

Reviewer #2 would also like us comment on the effect of the HFI on the spin diffusion length in organic spin valves. To our knowledge, the most direct evidence for spin relaxation from HFI was reported by Nguyen et al. (Nat. Mat. **9**, 345–352, 2010), a molecule with very weak SOC, where λ_S increased from 16 nm to 45 nm upon deuteration. Those magnetoresistance measurements were conducted at fields of 0 – 20 mT. However, we would like to stress that the impact of the HFI on spin relaxation is strongly reduced at higher magnetic fields.

On the other hand, spin-orbit coupling adds a term to the Hamiltonian that resembles an effective field $\mathbf{F}(t) = \mu_B \Delta g(t) \cdot \mathbf{B}$ which scales linearly with the external field \mathbf{B} (see line 288 of the manuscript). Therefore, SOC remains effective at flipping the spin even at higher fields.

We conducted our ESR measurements at fields of ~350 mT and found that at such field strengths, T_1 for isolated molecules is strongly dependent on the g-shift and therefore the effective SOC. The only exceptions in our series are molecules with negligible SOC and many hydrogens attached to the conjugated system such as rubrene. To prevent possible misunderstandings and to address the referee's concern, we inserted the following paragraph at line 302:

Note that fluctuating HFI fields create perturbations of the form $\mathbf{F}_n(t) = \mathbf{A}_n(t) \cdot \mathbf{I}_n$ where $\mathbf{A}_n(t)$ is the coupling tensor for the nuclear spin \mathbf{I}_n . This expression does not depend on the external field and the HFI therefore become less effective at flipping the spin when the Zeeman splitting increases at higher fields. In contrast, spin lattice relaxation by fluctuating SOC fields will remain equally effective at all fields. Measurements on organic spin valves for instance are typically performed at different magnetic fields between 5 - 500 mT, depending on the switching field of the ferromagnetic electrodes. The effect of HFI fields on spin relaxation will therefore be suppressed to different degrees which complicates a systematic comparison of spin diffusion lengths.

3) A similar topic has been studied by Drew et al. (PRL 110, 216602 (2013)) using a low energy muon spin rotation apparatus. The SOC strength was studied in solid state films rather than solutions. This is supposed to be more realistic than the study of SOC in solution. They found that the SOC strength does not depend on the molecular structures but the composition in the chemical backbone. This is in contradiction to the result obtained by the authors. I think due to the lack of the inter-molecular interaction, the isolated molecules in solution might have very different spin interaction in comparison to that in the solid state films. How would the authors design experiments to rule out this case?

Reviewer #2 suggests that a) the authors of Phys. Rev. Lett. (PRL) 110, 216602 (2013) find little or no dependence of SOC strength on the molecular geometry, but only on its composition, and b) that measurements in solid state thin-films are more realistic than our solution-based measurements.

In PRL 110, 216602 (2013), two distinct series of molecules based on the Alq3 and TES geometries, are studied with substitutions of increasing atomic weight. Drew's study was designed as precisely an attempt to eliminate structural effects on the SOI and just consider atomic composition. Drew's study is in fact not in contradiction to our work. It was simply not designed to investigate effects of molecular structure, whereas in our study we have included on purpose molecules with identical atomic composition, but different molecular structure.

In terms of the relevance of solution-based measurements we readily acknowledge that solution-based measurements can only be a first step to understanding mechanisms for spin relaxation in solid samples. However, the simplicity of the solution-based ESR measurements has allowed us to investigate a wide range of molecules that could practically not have been studied in solid-state experiments. This has allowed us to investigate both the effects of atomic composition (already reasonably well understood from the work of Drew and others) as well as the effects of molecular structure that had previously not been studied.

Furthermore, our discussion of the spin relaxation process in terms of time-varying spin-orbit fields in Section 3 suggests that similar physics is also likely to hold in solid systems: In particular, we expect that the relative

strengths of the spin-orbit fields between different molecules are likely to be broadly similar in solids and solutions. There will of course be differences, for example because the spin densities can be spread over multiple molecules by intermolecular interactions. However, our proposed framework of interpreting the strength of spin-orbit coupling in terms of the overlap between spin and orbital angular momentum distributions is likely to remain valid, i.e., we expect the linear relationship between atomic spin densities and the g-shift to still hold in thin films. Another difference in solids will be that the temporal fluctuations of the spin-orbit fields are not caused by molecular tumbling motions, as in solution, but by intermolecular vibrations and charge hopping that occur, however, on the same timescale.

For these reasons, we believe that first understanding spin relaxation phenomena in simple solution systems before investigating more complex solid systems is likely to be a fruitful scientific path. We emphasized this point in the Conclusion rewriting the last sentences:

In the solid state, the spin densities are likely to spread over multiple molecules but we expect Δg to still be measure of the effective SOC. Since the fluctuation amplitudes will still scale with the latter, knowing the relaxation times in solution should provide a good estimate for both the coherence and spin lattice relaxation times in thin films.

4) One of the most important issues in organic spintronics in these days is how to distinguish the SOC and HFI effect. The authors need to extract the HFI result from their data for the comparison.

We agree with reviewer #2 that the ability to distinguish effects of HFI and SOC on spin relaxation is of fundamental importance in organic spintronics. The remaining uncertainty about the magnitude of SOC in organic semiconductors was a main motivation for our study.

The most reliable and unambiguous way to distinguish HFI effects on T_1 two lies in deuteration and the resulting suppression of the HFI. Unfortunately, deuterated molecules are challenging to synthesize and therefore hard to obtain – in fact, we could only source d28-Rubrene out of all 11 studied molecules.

Consequently, we chose to study spin relaxation in a range of molecule were the effective SOC varies over a large range and can be precisely quantified. The first half of the manuscript therefore demonstrates how such a quantification is possible through the g-shift and the second half show that the spin lattice relaxation time indeed follows a clear $T_1 \sim (\Delta g)^{-2}$ dependence. We have therefore stressed this point in line 299 of the revised manuscript:

Fig. 3a shows that T_1 indeed follows this proportionality with a remarkable accuracy. We conclude that spin lattice relaxation is therefore dominated by the effective SOC at magnetic fields of ~ 350 mT.

Significant deviations from this scaling indicate contributions from HFI fields and are only present for rubrene and TIPS-pentacene in our series. From those two molecules, we can meaningfully estimate the contribution of HFI to the spin lattice relaxation. We therefore added the following section to the discussion of the HFI contribution to spin lattice relaxation (line 313 and following):

In a control experiment, we repeated the measurement on fully deuterated d28-rubrene, which strongly suppresses hyperfine interactions. Together with a reduction of the HFI couplings by a factor of 2-3 we observe an increase of T_1 by more than an order of magnitude which brings it in line with the other molecules.

The large deviation from the $T_1 \propto (\Delta g)^{-2}$ dependency can therefore be traced back to hyperfine interactions. We can quantify their contribution by summing over SOC and HFI relaxation rates $T^{-1} = T_{1,HFI}^{-1} + T_{1,SOC}^{-1}$ which yields pure HFI relaxation times of ~ 13 μ s for h28-rubrene and ~ 30 μ s for TIPS-pentacene. Such an estimate is only meaningful when the deviation from $T_1 \propto (\Delta g)^{-2}$ is larger than the measurement uncertainty, i.e., when the contribution of $T_{1,HFI}$ to T_1 is not negligible. We therefore cannot systematically extract HFI contributions for most of the molecules in this study but instead can identify a domain of weak SOC with $\Delta g < \sim 500$ ppm where, in the presence of a sufficient number of nuclear spins, hyperfine fields will significantly contribute to spin lattice relaxation.

We hope that this clarifies the scope and the limitations of our work.

Reviewer #3 (Remarks to the Author):

This manuscript reports a study into the tuning of spin-orbit interaction (SOI) in organic semiconductors. This is an area of wide interest, as pointed out by the authors, and where there has been significant experimental interest over the last 5-10 years. It should be stressed that much of the most frequently cited literature on spin interactions in organic semiconductor, particularly with respect to organic spin valves and organic magnetoconductance, has been relatively empirical and to an extent polarised with a significant community who have claimed the SOI are less important in organic semiconductors (due to the low mass of most of the constituent atoms). This manuscript provides a significant development in that it not only shows that not only can SOI be an important spin interaction mechanism but also highlights the circumstances under which hyperfine interactions can dominate. As such the work is a highly valuable contribution and helps to move the research field into a more balanced regime where the effect of the different contributions can be clearly assessed. The result of this is that the authors have been able to determine rules by which the effective spin-orbit coupling can be engineered through molecular design in order to achieve the desired properties for a range of spin based devices.

I strongly believe that this manuscript marks a significant development in our ability to control spin interactions in organic semiconductors and hence that it should be published in nature Communications.

We thank the reviewer for the positive assessment of our work and the recommendation for publication.

I do have a few relatively minor changes that I would like to see in order to improve the readability of the work for a broad audience. These are primarily on the definition of acronyms within the manuscript. As is common with many works on organic semiconductors there are a wide range of acronyms that are used for the molecules in order to make the work readable. With 32 different molecules this becomes an absolute necessity. However, not all of the acronyms are defined (e.g. DNTT and DATT on page 3, BSBS, DNAA on page 4). As it may make the reading of the manuscript difficult to follow if they are each spelt out in the text I would suggest that an expanded version of table 1 and S2 (to include all molecules, acronyms, molecular structure and chemical name) should be included in the supplementary information. This would particularly benefit the non-chemists reading the manuscript.

We understand the reviewer's concern regarding a clear definition of all acronyms – at times the non-chemist authors have been confused by the nomenclature as well.

We have therefore followed the reviewer's suggestion and expanded Tables 1 and S2 to include full chemical names and molecular structures in addition to the acronyms. In addition, we have also revised the paragraph introducing the various acronyms for a more structured introduction of the molecular geometries (lines 100 – 114 in the revised manuscript).

On a similar note the use of $D\{N,A\}\{TT,SS\}$ is “obvious” only after I've gone back through the whole manuscript with the sole purpose of trying to make sense of it.

We have expanded the brackets in line 169. The sentence now reads:

The effect is identical for the sulphur-based analogues, and similar but weaker in DNTT, DNSS, DATT and DASS.

Similarly although SOMO is well known by those in the field (Fig 1(a)) it should still be defined.

We thank reviewer #3 for the diligent reading of our manuscript, we indeed forgot to define this acronym. We added the definition of SOMO as “singularly occupied molecular orbital” to the caption of Fig. 1. We also moved the explanation of the charged induced UV-vis absorption spectra from the main text to the figure caption as it is not essential to the message of the paper.